

**Technical note: Rapid phase identification of apatite and zircon grains for geochronology**
**using X-ray micro-computed tomography**
Emily H. G. Cooperdock[1]*, Florian Hofmann[1,2]*, Ryley M. Collins[1], Anahi Carrera[1], Aya
Takase[3], and Aaron J. Celestian[4]
[1]University of Southern California, Department of Earth Sciences, 3651 Trousdale Parkway, Los
Angeles, CA 90089, USA
[2]University of Alaska Fairbanks, Geophysical Institute, 900 Yukon Dr, Fairbanks, AK 99775,
USA
[3]Rigaku Americas Corporation, 9009 New Trails Drive, The Woodlands, TX 77381, USA
[4]Mineral Sciences Department, Natural History Museum of Los Angeles County, 900 Exposition
Boulevard, Los Angeles, California 90007, USA
*Authors contributed equally to this work.
*Correspondence to*: cooperdo@usc.edu



**Abstract**

Apatite and zircon are among the best-studied and most widely used accessory minerals
for geochronology and thermochronology. Given that apatite and zircon are often present in the
same lithologies, distinguishing the two phases in crushed mineral separates is a common
challenge that many laboratories face. Here we present a method for efficient and accurate
apatite and zircon mineral phase identification using X-ray micro-computed tomography
(microCT) of grain mounts that provides additional 3-dimensional grain size, shape, and
inclusion suite information. In this study, we analyzed apatite and zircon grains from Fish
Canyon Tuff samples that underwent methylene iodide (MEI) and lithium heteropolytungstate
(LST) heavy liquids density separations. We validate the microCT results using known standards
and phase identification with Raman spectroscopy demonstrating that apatite and zircon are
distinguishable from each other and other common phases, e.g., titanite, based on microCT X-ray
density. We present recommended microCT scanning protocols after systematically testing the
effects of different scanning parameters and sample positions. This methodology can help to
reduce time spent performing density separations with highly toxic chemicals and visually
inspecting grains under a light microscope, and the improved mineral identification and
characterization can make geochronologic data more robust.

**1 Introduction**


Apatite and zircon are mineral phases widely used for geochronology and
thermochronology using the U-Pb (e.g., Bowring and Schmitz, 2003), (U-Th)/He (e.g. Farley,
2002), and fission track (e.g. Tagami and O'Sullivan, 2005) methods. Correct identification of
these phases (e.g. Guenthner et al., 2016), characterization of the crystal shape (Farley et al.,
1996), and the absence of mineral and fluid inclusions (e.g. Lippolt et al., 1994; Vermeesch et
al., 2007) are important factors in producing reliable, high-quality geochronologic data. The
standard approach to selecting apatite and zircon grains for geochronology is to 1) crush and
grind rock samples into their mineral constituents, 2) perform magnetic and density separation
using a Frantz isodynamic separator and heavy liquids to filter for the mineral of choice, and
then 3) pick individual grains from these separates under a transmitted light microscope
(Gautheron et al., 2021).



Different heavy liquid solutions used for density separation can either produce grain
fractions that have apatite and zircon mixed together or separated (e.g., Dumitru and Stockli,
1998; Koroznikova et al., 2008). The density of apatite ($Ca_5(PO_4)_3(F,OH,Cl)$) is 3.10-3.25 $g/cm^3$
and depends on the solid solution composition between fluorapatite, chlorapatite, and
hydroxylapatite (Hughes et al., 1989). Zircon ($ZrSiO_4$) can display densities between 3.9 and 4.7
$g/cm^3$, depending on the degree of metamictization (Holland and Gottfried, 1955). Although
density-separated apatite and zircon fractions make picking the correct mineral easier (Dumitru
and Stockli, 1998), the process often includes the use of toxic halogenated organic solutions,
such as bromoform ($CHBr_3$) and diiodomethane ($CH_2I_2$, methylene iodide, commonly
abbreviated as MEI, MI, or DIM; e.g. Hauff and Airey, 1980). Typically, bromoform (2.89
$g/cm^3$) is used in a first step to separate quartz and feldspar and the resulting heavy fraction is
then treated with MEI (3.32 $g/cm^3$) to separate apatite and zircon (e.g. Dumitru and Stockli,

56    1998).

Both Bromoform and MEI are known to be toxic. Specifically, MEI can cause acute
symptoms through skin contact or inhalation, and acute toxicity and death have been documented
for a case of ingestion (Weimerskirch et al., 1990). MEI has also been shown to be mutagenic
meaning acute or long-term exposure may impact reproductive health, particularly in pregnant
women (Van Bladeren et al., 1980; Osterman-Golkar et al., 1983; Buijs et al., 1984; Roldán-
Arjona and Pueyo, 1993). In addition, samples separated with MEI are typically washed with
acetone, and the mixture of these chemicals is highly flammable. Burning MEI has the potential
to produce large amounts of free iodine, which also poses a significant health risk (Hauff and
Airey, 1980). Due to its toxicity, MEI must be used in a vent hood with proper personal
protective equipment (PPE) and requires special training in safe handling techniques (Dumitru
and Stockli, 1998).
Safety precautions required for hazardous chemical handling may exclude workers or
students with conditions that do not allow them to comply with the safety precautions. For
example, personal protective equipment may only be available in restricted sizes, and fume hood
design is often incompatible with the use of wheelchairs or other mobility devices. Thus,
eliminating hazardous chemicals from laboratory procedures results in both a safer work
environment and a more inclusive workplace.



Many labs elect to use lithium heteropolytungstate (LST), lithium metatungstate (LMT),
and sodium polytungstate (SPT) because they are generally non-toxic and relatively inert
(Munsterman and Kerstholt, 1996; Mountenay, 2011). Similar to bromoform (but less toxic)
these heavy liquids can be used at densities of 2.8-3.0 g/cm$^3$ to remove quartz and feldspar from
the separate, but they do not separate apatite from zircon. Zircon and apatite crystals in natural
samples exhibit a wide variety of morphologies depending on the sample history and can be
difficult to distinguish by eye under a binocular microscope despite the fact that they have
different compositions and crystal structures. Optical methods such as crossed polarizers are
often used in addition to crystal shape to distinguish these phases from each other as well as from
other phases such as titanite, xenotime, monazite, allanite, rutile, baddeleyite, etc., but are not
always able to uniquely identify the phase of a particular grain.
Mistaken mineral identification can lead to significant issues in data analysis, quality, and
interpretation. Depending on the geochronologic technique employed, this misidentification
might be detected further along in the analytical procedures. In (U-Th)/He analysis, a mistake
may be realized during degassing or dissolution. Due to their differential diffusion behavior,
zircon usually requires higher temperatures and longer laser-heating times to fully extract He
than for apatite (e.g. Farley, 2002). Apatite dissolves readily in a weak nitric acid, whereas zircon
needs to be subjected to extensive Parr bomb pressure dissolution procedures using a mixture of
nitric acid, hydrochloric acid, and hydrofluoric acid to be completely dissolved (Farley, 2002).
As a result, a misidentified mineral may not be completely degassed or dissolved during the
analytical procedure, leading to erroneous results. The presence of Ca or Zr in dissolved mineral
solutions can be used during subsequent isotope-dilution ICP-MS analysis to test whether the
correct phase was chosen for the analysis, as was demonstrated for (U-Th)/He by Guenthner et
al. (2016).
Similar issues arise in other methods. In laser ablation analysis as part of U-Pb or (U-
Th)/He dating, the ablation characteristics and the presence of Ca or Zr in the analyte can be used
as diagnostic criteria. Etching parameters for fission track, such as the type and molarity of acids,
etching time, and temperature conditions, are highly phase-specific and need to be tightly
controlled to yield reproducible and internally consistent data (Tagami and O'Sullivan, 2005).
Applying zircon etching procedures to apatite grains might lead to the complete loss of a sample.



Given the amount of time and materials required by these analytical methods,
misidentification of minerals can lead to significant monetary and time-effort losses. Therefore,
an efficient pre-screening technique to confirm apatite and zircon phases for geochronologic and
thermochronologic application can help to avoid unsuccessful partial analyses of misidentified
samples and lead to more robust and reproducible data. Many laboratories have developed
techniques to reduce mineral misidentification. These can include having a more experienced
user look over selected grains, analyzing pre-selected grains under a scanning electron
microscope (SEM) to measure elemental compositions with energy dispersive spectroscopy
(EDS), using Raman spectroscopy for phase identification, and others.
We test whether X-Ray micro-computed tomography (microCT) scanning can be used as
an effective pre-screening tool to distinguish between apatite and zircon and to detect
misidentification of grains. The difference in apatite and zircon composition and densities (3.1-
3.2 g/cm$^3$ and 3.9-4.7 g/cm$^3$, respectively) lead to differential X-ray absorption, which yields
characteristic grayscale value contrast in microCT data (e.g. Ketcham and Carlson, 2001). In
addition to phase identification, microCT data yields high-resolution 3-dimensional grain shape
measurements and reveals internal heterogeneities, such as fractures or inclusions (Evans et al.,
2008; Glotzbach et al., 2019; Cooperdock et al., 2019). Resolution varies by instrument and
acquisition parameters; the instrument used in this study achieves a maximum voxel resolution
of ~2 µm/10 µm$^3$. We explore different acquisition parameters to optimize the distinction
between different minerals and minimize the scan time to yield a streamlined procedure for
routine pre-screening of mineral grains for geochronologic applications.
**2 Materials and methods**
**2.1 Mineral separation**
We selected Fish Canyon Tuff (FCT) as a test sample because it contains both apatite and
zircon and is used as an age standard in many applications of geo- and thermochronology
(McDowell et al., 2005; Donelick et al., 2005). We obtained three separate FCT samples: one
mineral separate of a MEI heavy fraction given to us by the UTChron Laboratory at the
University of Texas at Austin (UT-FCT), and two that we collected from two FCT localities near
Monte Vista, CO (USC-FCT1: 37°36'38.73" N, 106°42'19.93" W; USC-FCT2: 37°38'22.21" N,





106°17'57.77" W). The two whole-rock samples were crushed on a jaw crusher and disk mill at
the University of Southern California. Crushed samples were sieved and the 75-250 µm size
fraction was washed before using a hand magnet and a Frantz isodynamic magnetic separator to
remove magnetic fractions. Samples then underwent density separation using lithium
heteropolytungstate (LST). This is a water-based, low-toxicity heavy liquid with a maximum
density of 2.85 g/cm$^3$ at room temperature that produces a heavy mineral separate with apatite
and zircon (and other phases) mixed together. Sample types and names are summarized in Table

1.

The UT-FCT separate supplied by the University of Texas at Austin was processed using

the same mineral separations procedures with the following exceptions: the samples were density
separated on a Gemeni water table prior to magnetic separation, and the sample experienced a
two-step heavy liquids separation using bromoform and MEI. These heavy liquids are more toxic
than LST but have densities of 2.95 g/cm$^3$ and 3.32 g/cm$^3$, respectively, and should yield grain
fractions that separate apatite from zircon. Only the MEI heavy fraction was used for this
experiment.

As a reference for microCT imaging, we used mineral standards for apatite, zircon, and

titanite from existing collections. Two Durango apatite standards from large apatite crystals were
supplied by the UTChron laboratory at the University of Texas at Austin (UT-DUR) and Caltech
(ClT-DUR). We used shards from large crystals of Sri Lankan zircon (SL1) from Caltech (Farley
et al., 2020) and Minas Gerais titanite (MG1) from the Natural History Museum of Los Angeles
County (more specific sample location information is not known). These standard crystals were
gently hand crushed and sieved to <75 µm, 75-250 µm, and >250 µm size fractions.

Table 1. Mineral standards and unknowns used in this study. Large standard crystals were
crushed to obtain shards to be used as a reference for microCT analyses. Unknown grains were
extracted from FCT whole-rock samples.

| Sample | Minerals | Type | Grain type | Sample Name | Density Separation |
|--------|----------|------|------------|-------------|---------------------|
| UT-DUR | Apatite | Standard | Shard | Durango | none |
| ClT-DUR | Apatite | Standard | Shard | Durango | none |
| SL1 | Zircon | Standard | Shard | Sri Lanka | none |



| Sample | Minerals | Type | Grain type | Sample Name | Density Separation |
|---|---|---|---|---|---|
| MG1 | Titanite | Standard | Shard | Minas Gerais | none |
| UT-FCT | Apatite, Zircon | Unknown | Grain | Fish Canyon Tuff | bromoform, MEI |
| USC-FCT1 | Apatite, Zircon, Titanite | Unknown | Grain | Fish Canyon Tuff | LST |
| USC-FCT2 | Apatite, Zircon, Titanite | Unknown | Grain | Fish Canyon Tuff | LST |


## 2.2 Making crystal mounts

Graduate students were tasked with picking mineral grains that looked like apatite or zircon and covered a range of grain sizes and morphologies from the three FCT samples using a Nikon SMZ25 optical microscope. It is notable that all samples, including the MEI separate, yielded both apatite and zircon. The selected grains were placed onto grain mounts for microCT analysis (see Sect. 2.3). Each mount also included known mineral standards for reference and normalization (Fig. 1a). Three grain mounts were constructed (Mount A, B, and C, see Fig. 2). Mount A included 36 grains from UT-FCT "unknowns," 10 shards of SL1 zircon, and 15 shards of ClT-DUR apatite. Mount B included 39 grains of USC-FCT1 "unknowns," 32 grains of USC-FCT2 "unknowns," 9 shards of SL1 zircon, and 24 shards of UT-DUR apatite. Mount C included 11 shards of SL1 zircon, 15 shards of CIT-DUR apatite, and 15 shards of MG1 titanite standards. We used the 75-250 µm size fraction and >250 µm size fractions of the mineral standards to test the impact of grain size on grayscale values in microCT data. On Mount C, individual shards from each mineral were distributed evenly across the mount to test whether there is any spatial variability in X-ray attenuation and grayscale.

We assembled grain mounts by cutting small plastic shapes (rectangles, squares, or circles) out of 1 mm thick plastic slides and placing double-sided adhesive tape on one side. Mounts for vertical scans (when the mount is standing upright on the top of the sample holder) were constructed by cutting ~3 mm by 4 mm rectangles from plastic slides of 1 mm thickness, which was covered with double-sided adhesive tape. Grains were placed on the upper part of the rectangle mount (Fig. 1a), and the end without grains was inserted into dental wax to hold the



mount in place, vertically, on top of the sample holder (Fig. 1b). We tested different brands of
double-sided adhesive tape and found that some brands appear clear under a transmitted light
microscope while others have significant interference colors and visible fibers. Double-sided
tape selection did not affect microCT data.
Prior to placing the grains, the plastic mounts were temporarily secured to a glass slide
with double-sided tape to hold them in place. Individual crystals were selected from mineral
separates and placed on the tape using tweezers and needles under a light microscope. Grains
were spaced to avoid touching, with up to 104 total crystals per mount. Optical micrographs of
the mount and each individual crystal were taken with transmitted and reflected light as well as
with crossed polarizers.


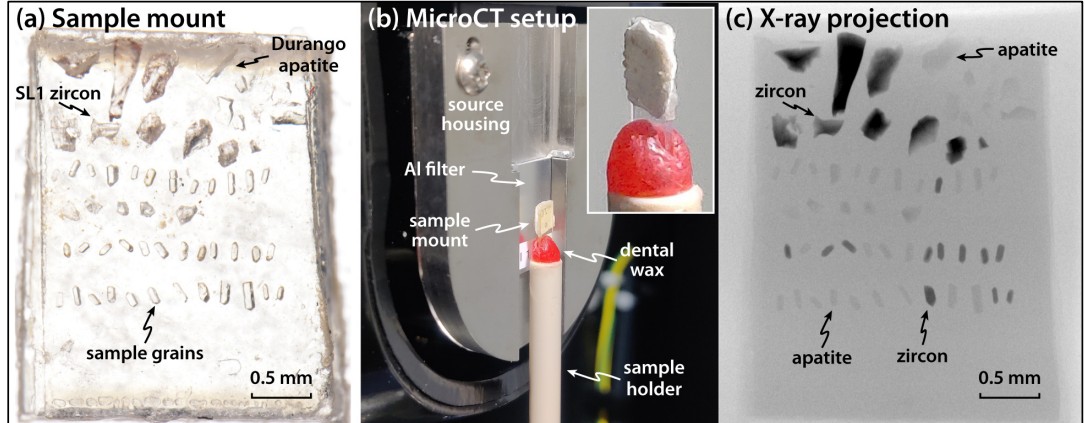

Figure 1: (a) Transmitted light micrograph of a sample mount with known apatite and zircon
standard shards and unknown sample grains made from a plastic slide and double-sided tape,
about 3 mm in width. (b) Sample mount installed vertically in the microCT instrument secured
on top of a sample holder with dental wax. Insert shows a closer view of the sample mount in
measurement position. (c) X-ray projection of the same mount as in (a). Zircon grains show up as
darker (more X-ray absorption) than apatite grains. The brightness in projections is controlled by
the material-specific X-ray attenuation as well as by the integrated thickness of the traversed
material.





**2.3 MicroCT scanning**

All microCT scans were acquired on a Rigaku CT Lab HX130 benchtop microCT instrument at the USCHelium Laboratory at the University of Southern California. Individual mounts were installed vertically (perpendicular to the X-ray beam direction, parallel to the detector plane; see Fig. 1b) in order to minimize the effect of interference from X-ray artifacts such as shadowing between individual grains due to beam hardening and photon starvation (see Section 3.2 and Fig. 7). Mounts were scanned at accelerating voltages of 130 and 60 kV with currents of 61 and 133 µA, respectively. We used a 1.0 mm thick aluminum filter to selectively remove lower energies from the polychromatic beam in order to reduce the effect of beam hardening (see Hanna and Ketcham, 2017, for details). Total instrument run times were between 18 seconds and 125 minutes using continuous and step scanning with a field of view (FOV) of 5 mm diameter and 3.8 mm height (see Table 2). Continuous scans were done for 18 s, 4 min, 17 min, and 68 min. Over this time, the sample is rotated and X-rays are continuously counted on the detector. We also performed 125 minute step scans (500 ms exposure time, 1500 projections, 4 integrations), in which the sample is rotated in steps and the detector moves between the steps to reduce ring artifacts. As a result, the 125 minute scan time includes 50 minutes of actual X-ray exposure and 75 minutes of instrument adjustment. Note, in continuous scans the scan time and exposure time are the same because there is no detector adjustment. We report the total instrument scan time in Table 2 and the total exposure time on Figure 7. Reconstructions were computed using the Rigaku CT Reconstruction software. Continuous scans were reconstructed to yield volumes with a width and length of 1024 voxels. Step scans were integrated for longer times than the continuous scans and yielded enough data to be processed at full resolution (width/length of 2784 voxels) while maintaining a usable signal-to-noise ratio.

Table 2. Scan parameters tested in this study.

| Scan voltage (kV) | Scan type | Total scan times (minutes) | Voxel size (µm) | Volume size (pixels) | File size (GB) |
|---|---|---|---|---|---|
| 60 and 130 | continuous | 0.3, 4, 17, 68 | 5.7 | 1024x1024x708 | 1.4 (0.2 cropped) |
| 60 and 130 | step | 125 | 2.1 | 2784x2784x1931 | 27.8 (2.4 cropped) |



### 2.4 MicroCT data analysis


The reconstructed microCT data was processed with Dragonfly (Version 2021.1) by

Object Research Systems. Reconstructed volumes of each mount with all different scan times
and X-ray energies were loaded into Dragonfly. The volumes scanned at 60 kV for 68 min were
used as a reference since they displayed the best signal-to-noise ratio of all the tested scan
parameters. Volumes were registered relative to the 60 kV/68 min scans using the Image
Registration tool, which translates and rotates volumes to align scans. Grains were segmented in
the 60 kV/68 min scan volumes by creating regions of interest (ROI) using histographic
segmentation, which delineates grains from their surroundings (air or adhesive tape) based on
threshold grayscale values. The resulting volumes were filtered by applying a 3D opening
operation (a combination of erosion and dilation which removes small objects, like dust, while
not changing the geometry of large volumes) and eroded by one voxel to remove the effect of
rapid changes in grayscale value near the grain boundary.

Each grain was separated into an 'object' by creating a Multi-ROI (a ROI that contains

multiple objects) from continuous segments in which voxels are connected by at least one of
their faces (6-connected). Each grain 'object' consists of hundreds to thousands of voxels that
can be used to calculate grayscale statistics. Small fragments separated from larger grains of less
than 100 voxels were not used for further analysis to ensure the measurements have statistical
significance. In this way, individual grains were mapped out and distinguished from other small
objects in the scan (e.g., chipped pieces or detritus on the adhesive tape). The geometry of the
segmented objects was resampled to fit each volume, and information on the position, size,
surface area, and greyscale value distribution of each grain was extracted from the Multi-ROIs.

Absolute grayscale values can change between scans since they are dependent on the

scan geometry, acquisition parameters, arrangement of grains, and processing, with internal
normalization and scaling being applied during reconstruction. To make scans comparable, we
chose to normalize the grayscale values of all grains on a mount by the average grayscale value
of the SL1 zircon grains in the same volume. We also computed the ratio of the grayscale values
of the 60 kV and 130 kV scans with otherwise identical scan parameters to yield a dual-energy
parameter.





**2.5 Phase validation by Raman spectroscopy**

To validate the different phases observed in microCT data, we determined the mineral

phase of 35 grains in Mounts A and B by Raman spectroscopy. This included a subset of 28
unknown grains from FCT samples and 7 shards of known mineral standards (Fig. 2).
Representative grains were selected to encompass a range of grain sizes and morphologies,
positions on the mount, and microCT grayscale contrast. After microCT scanning, the grain
mounts were transferred to a glass slide, and grains were analyzed using a HORIBA XploRA
PLUS spectrometer at the Natural History Museum of Los Angeles County. Apatite, zircon, and
titanite were identified by matching baseline-corrected spectra with comparison spectra from the
RRUFF database (Lafuente et al., 2005) using CrystalSleuth. Raman spectral analyses were
conducted using a green 532 nm diode laser at 50% laser power, a diffraction grating of 1880
gr/mm, a 100x (0.9 NA) objective, 200 μm slit, and 300 μm pinhole for confocal optical
geometry. Raman spectra were collected in the range of 100-1600 cm$^{-1}$ with each grain analyzed
with a 3 s exposure averaged from 10 acquisitions.





Figure 2: Transmitted light micrographs (a,d,g), microCT slices (b,e,h), and microCT volume renderings (c,f,i) of Mounts A, B, and C. MicroCT slices show a large contrast between apatite/titanite (darker) and zircon grains (brighter). Grayscale color and grain relief in 3D renderings are distinct for different mineral phases. The 3D renderings show Raman-validated



grains highlighted and known standard shards circled in blue (apatite), green (zircon), and
titanite (orange). Baseline-corrected Raman spectra of representative grains and reference spectra
from the RRUFF database (including record numbers) are shown below the images. Numbers in
circles indicate the grains in the volume renderings which correspond to the sample Raman
spectra.

## 3 Results and discussion

Different microCT scanning parameters were systematically tested on the same three
grain mounts to determine the optimal scan conditions for distinguishing between mineral phases
while minimizing cost, time, and data file sizes. Individual microCT data file sizes range from 2
to 28 GB depending on acquisition and processing parameters. Reconstructing and manipulating
large datasets can require specialized computers with demanding system requirements for data
storage, memory, and processing power. The microCT data for single grain mounts, like the ones
used in this study, can be cropped to produce manageable file sizes that can be viewed and
analyzed without the need for specialized computers. We determined that for the instrument used
here a continuous scan time of 17 min at 60 kV (5.7 µm resolution) is sufficient for mineral
identification between apatite and zircon. For phase identification plus high-resolution surface
area and volume for 3D grain geometry measurements (as is typical for (U-Th)/He
thermochronology), we recommend using a 125 min step scan at 60 kV (2.1 µm resolution).
These parameters are optimized for apatite and zircon and can be modified for other minerals of
interest. Below, we evaluate the effects of X-ray energy, grain size, and spatial distribution on
quantitatively distinguishing zircon from apatite using microCT data.

### 3.1 Theoretical X-ray attenuation

We calculated the theoretical X-ray total attenuation coefficients of apatite, zircon,
titanite, monazite, and rutile (Fig. 3a) for a range of X-ray energies commonly used for microCT
(~30-230 keV) using MuCalc (https://www.ctlab.geo.utexas.edu/software/mucalctool/), a
Microsoft Excel plugin which uses data from the NIST XCOM database of mineral-specific
parameters (Hanna and Ketcham, 2017). The modeled attenuation coefficients predict how X-
rays interact with different minerals. The greater the difference in attenuation coefficients, the
more distinct two mineral phases will appear in microCT data.





Based on these calculations, zircon has a much higher attenuation coefficient than apatite
across the energy spectrum. At lower energies, the difference between the attenuation
coefficients of other minerals relative to zircon (Fig. 3b) is greater than at higher energies. The
attenuation coefficients of apatite, zircon, titanite, and rutile converge around 200-300 keV.
Thus, energies less than ~200 keV should make zircon grayscale values distinguishable from
apatite and other lower attenuation phases (i.e., zircon appears brighter in reconstructed microCT
data as seen in Fig. 2). The attenuation coefficients of apatite and titanite are similar at all
energies, but display slightly more divergence <80 keV. The observed X-ray attenuation of
actual mineral grains might differ from these predictions due to material inhomogeneity,
compositional variation (such as endmember mixing and elemental substitution), crystal defects
(e.g., metamictization), inclusions, and artifacts due to shadowing from neighboring grains
(photon starvation) and beam hardening. In this study, we analyzed our mounts at the maximum
achievable voltage on the Rigaku CT Lab HX130 of 130 kV as well as a reduced voltage of 60
kV. These parameters may vary for other microCT instruments.


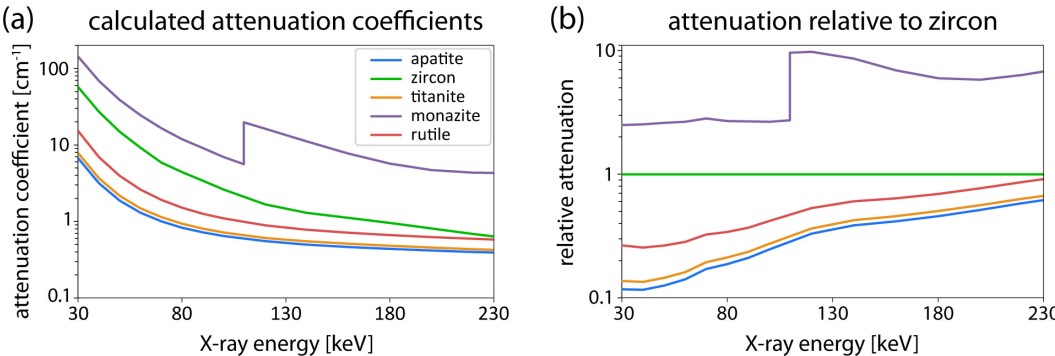


Figure 3: (a) Attenuation coefficients for commonly dated minerals over a range of X-ray
energies calculated with MuCalc. (b) The same modeled attenuation coefficient data normalized
by zircon. Generally, higher attenuation coefficients mean brighter grayscale values in
reconstructed microCT data. A greater difference in attenuation coefficients between mineral
phases aids in mineral identification.





### 3.2 Normalized grayscale values of grains


We use the 68-minute continuous scans to assess how grayscale values of individual
grains (or shards) vary at different scan energies and for different mineral phases. Grayscale
values for individual grains of unknowns and standards were normalized by the average value of
the SL1 zircon shards on each mount for each set of scan parameters. The absolute grayscale
value in the volumes depends on scanning conditions and reconstruction settings, thus internal
normalization makes the results comparable and independent of these parameters.
We found that apatite grains have grayscale values of about 22% and 27% (at 60 kV and
130 kV, respectively) of those of zircon grains (Fig. 4). The distributions are broad due to intra-
grain, inter-grain, and inter-sample variability, but the apatite and zircon populations are distinct
from each other so that individual grains can be uniquely identified. This also confirms the
theoretical modeling (Fig. 3) and the observations of different X-ray attenuation of apatite and
zircon grains in the X-ray projections (Fig. 1). The grayscale value distribution of titanite
overlaps partially with that of apatite and is sample-dependent, making a phase distinction
possible for some but not all grains. For example, the MG1 titanite mineral standard more
closely overlaps the apatite grains than the "unknown" titanite crystals picked from USC-FCT1
and 2, which are systematically slightly brighter (Fig. 5).
The separation between all of the distributions is greater for 60 kV than for 130 kV, as
predicted by the theoretical modeling above (Fig. 4). Therefore, volumes from scans at 60 kV
can be used to resolve smaller differences in X-ray attenuation than at 130 kV, which does not
have a pronounced effect on the apatite-zircon distinction but can be useful when trying to
distinguish between apatite and titanite. However, lower energy X-rays are less penetrating and
lead to more artifacts and noise in the resulting reconstructed data (Hanna and Ketcham, 2017).
Therefore, there is a trade-off between the absolute separation of phases in grayscale-value space
and the signal-to-noise ratio, the latter of which can be improved by longer scan times.

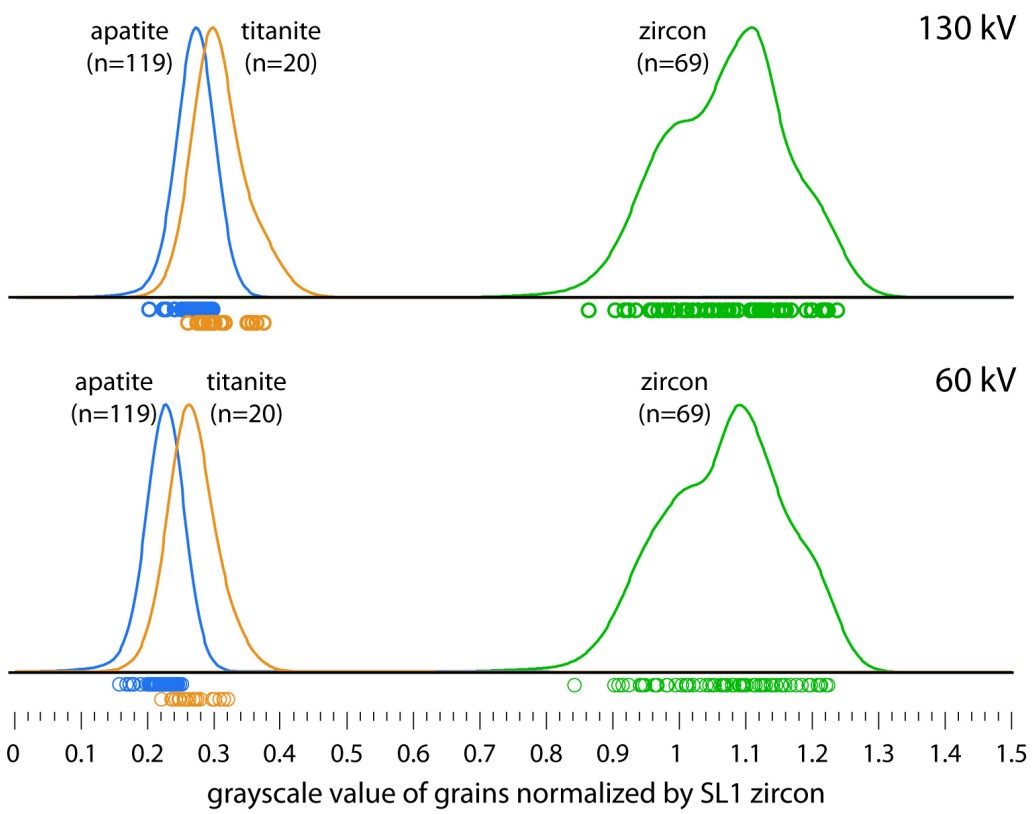


Figure 4: Kernel density estimates (KDEs) of all apatite, zircon, and titanite grayscale value
measurements (including standards) for 68 min scans calculated with an adaptive bandwidth
equal to the standard deviation of grayscale variation within each grain. Each KDE is an
aggregation of data from three different sample mounts and shows all individual data points. The
grayscale value of each grain was normalized by the average grayscale value of SL1 zircon
grains in the same volume. The difference between the attenuation of the three minerals is
greater at 60 kV than at 130 kV, as theoretically predicted.

We observed good reproducibility for average normalized grayscale values of
populations of the same sample across the three mounts (Fig. 5). For example, the average
normalized grayscale values of Durango apatite shards (UT-DUR) are all within uncertainty at
$0.255\pm0.046$ ($2\sigma$) for Mount A, $0.267\pm0.016$ for Mount B, and $0.272\pm0.014$ for Mount C. Some
of these average values are skewed by individual outliers, which are likely due to grain size
effects (see Section 3.4).





Although average grayscale values across grain populations are reproducible, we observe
a range of grayscale values for individual replicate grains from the same sample or of shards
from the same crystal (Fig. 5). This may be due to differences in bulk composition and structure.
For example, natural apatites are solid solutions of three different endmembers which have
different densities. The exact composition of any apatite grain will have an impact on its X-ray
absorption and hence the observed grayscale value. Zircon density is mainly controlled by
radiation damage (Holland and Gottfried, 1955), which can cause different densities for different
grains or of parts of the crystal in the case of pronounced zoning of radioactive elements. The
effect of differing grayscale values between different samples is most pronounced between the
titanite standard in Mount C and the titanite from FCT samples in Mount B (see Fig. 5). The
density of titanite has also been shown to be a function of crystal damage (Vance and Metson,

1985).

We segmented grains based on their outer surface and calculated the average grayscale
value of the material enclosed by that surface. It is necessary to exclude the outermost grain
boundary because it commonly appears falsely brighter due to beam hardening. However, if
there is internal heterogeneity, such as inclusions with higher or lower grayscale values, the
observed average grayscale value of any particular grain can be affected (expressed as RSDs).
Grains with a large fraction of inclusions of a particular type can therefore change the average
grayscale value and might lead to misidentification. One strategy to mitigate this would be to
filter certain histographic ranges of values within the segmented grains to exclude inclusions and
measure only the average grayscale value of the host grain. Alternatively, this could also be used
as a tool to identify individual crystals with inclusions, which would display higher or lower
average grayscale values than the rest of the population.
The grayscale value distribution within a particular mineral grain is dependent on the
natural variation of density and composition (such as zoning) as well as measurement noise. The
absolute $2\sigma$-variability of apatite and titanite grains is about 0.01-0.02 for apatite and 0.1-0.2 for
60 kV/68 min scans normalized by SL1 zircon (Fig. 5). In relative terms, this is a  5-10%
variation for apatite and titanite, and a 10-20% variation for zircon. Measurement noise in the
reconstructions is likely not the main contributing factor to this variation in the 68 min scans (see
Section 3.4 and Fig. 7). The remaining variations can be due to changes in material parameters
across a grain, inclusions of different densities than the host phase, and beam hardening. Overall,





the normalized grayscale value can be used to distinguish apatite and zircon, and to some extent
other phases such as titanite. Employing strategies to minimize noise and artifacts is important to
make this distinction robust for every analyzed mineral grain.

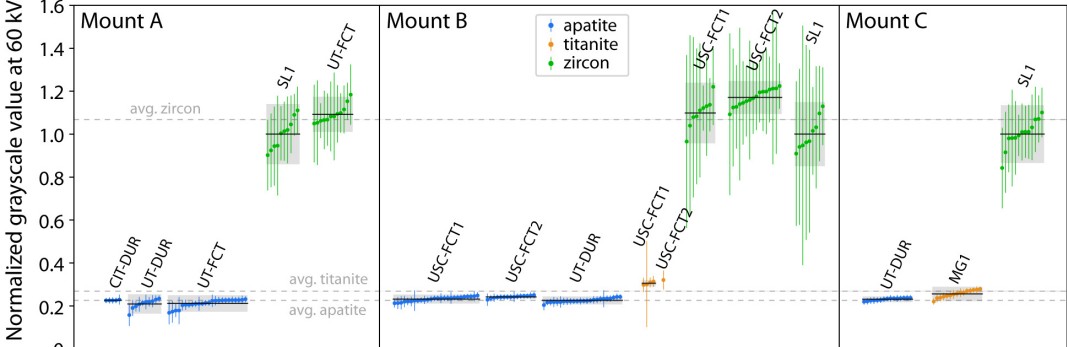

Figure 5: Mean grayscale values (normalized by SL1 zircon) for all grains measured in 60 kV/68
min scans, given with 2σ-variability and organized by mount and sample. Zircon is shown in
green, apatite in blue, and titanite in orange, as in the other figures. The average for each sample
is given as a black bar with the 2σ-variability shaded in gray. Averages for the whole populations
of apatites, zircons, and titanites are given as gray dashed lines. Zircon and apatite populations
for all mounts are distinct, while apatite and titanite populations show some overlap. There is
observable inter-sample variability in the mean normalized grayscale value of each mineral but
values for the same samples (e.g., UT-DUR) are reproducible within error between mounts.

## 3.3 Use of dual-energy data

The change of the attenuation coefficient with X-ray energy is a function of material
density and composition, and is characteristic for each mineral (Alves et al., 2014). Therefore,
the ratio of the attenuation at two different X-ray energies can be used as an additional parameter
to identify the mineral phase of a grain (e.g. Hanna and Ketcham, 2017). We observed a clear
distinction between apatite and zircon in this parameter as well (Fig. 6a). Titanite again appears
similar to apatite, but the separation between the two distributions is greater in dual-energy space
than in the 60 kV or 130 kV data alone. Therefore, this dual-energy parameter can be used as an





additional tool to distinguish phases that have similar absolute attenuation coefficients, and
hence appear similar in terms of grayscale values. This necessitates two scans of the same mount
at two different energies, as well as additional processing to align the two scans and compute
average grayscale values for both scans. However, the resulting data can be used to map regions
in dual-energy vs. single-energy plots (Fig. 6b), yielding a more robust phase identification for
individual grains.


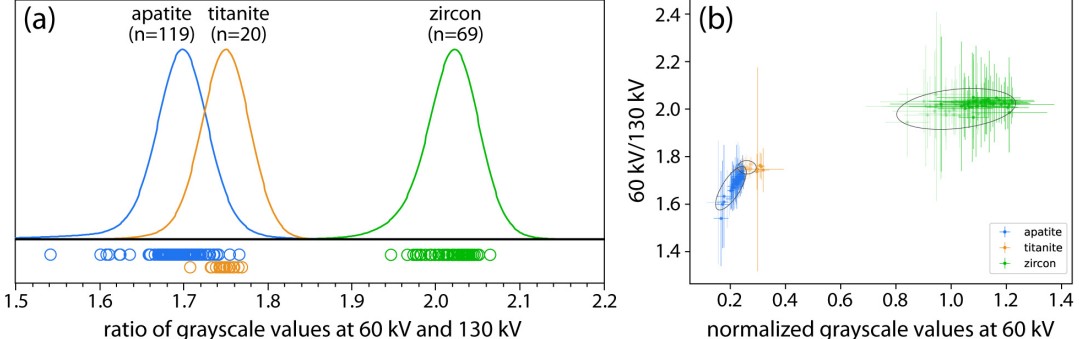


Figure 6: (a) Kernel density estimates of the ratios of the grayscale values at 60 kV and 130 kV
for grains from all three mounts. The mounts were scanned at 60 kV and 130 kV with otherwise
identical scan parameters and the grayscale values were measured at the same positions. Zircon
and apatite form very distinct distributions, and the populations of apatite and titanite overlap but
show more separation than grayscale values from scans at a single energy. (b) Dual-energy
parameters plotted against normalized grayscale values at 60 kV. Known standards are shown in
lighter colors and black lines outline the field of values of standards. Unknown sample grains of
apatites and zircons fall almost entirely within the field of standards. Titanite sample grains are a
significantly different brightness (grayscale values) than sample grains but have the same dual-
energy parameter.

### 3.4 Optimizing mount geometry and scan parameters

We tested the grayscale variability introduced by grain size, spatial distribution of the

grains on a mount, and direction of the mount during microCT data acquisition. Each of these



factors can affect the path that X-rays take through the grains and the preferential attenuation of
parts of the X-ray spectrum of a polychromatic beam (beam hardening), which can result in
artifacts that cause changes of the average grayscale for a given grain unrelated to the actual
mineral-specific X-ray attenuation. We found that image quality and signal-to-noise ratio
improved with increased scan time (Fig. 7), as is expected based on counting statistics. We
quantified variability in our data by calculating the relative standard deviation (RSD) of
grayscale value within each segmented grain, which is a measure of both natural variability of
the material and any superimposed measurement noise.

A clear distinction between apatite and zircon can already be observed in the 18 s scans

(Fig. 7), although the RSDs are high (0.2-0.3) for both apatite and zircon grains. The RSDs
decline with increasing scan time for otherwise constant experimental conditions (Fig. 6),
asymptotically approaching ~0.04 for apatite and ~0.08 for zircon. The remaining RSDs might
reflect the true natural variability of material parameters (density, endmember mixing, crystal
damage, elemental substitution, inclusions) within the mineral grains. For the particular
instrument and experimental setup employed here, the signal-to-noise ratio did not improve
significantly beyond a scan time of 17 min at a reduced resolution (voxel size of 5.7 μm). For
full-resolution reconstructions, a 125 min scan time was sufficient to produce comparable RSDs,
while also allowing for a smaller voxel size (2.1 μm) which is preferable for obtaining geometric
parameters, such as crystal size and shape for FT-corrections (Evans et al., 2008).

We also found that the orientation of the mount during data acquisition has a significant

effect on the data quality. A vertical orientation, perpendicular to the source and parallel to the
detector plane, produced much lower RSDs for the same scan conditions than a horizontal
position (Fig. 8). Highly attenuating phases (such as zircon) produce artifacts such as shadowing
and streaking (e.g. Hanna and Ketcham, 2017). When these artifacts overlap with other sample
grains, they can significantly alter the observed grayscale value of parts of grains which does not
reflect their actual X-ray attenuation and leads to erroneous measurements with increased RSDs
(Fig. 8). X-rays passing through a horizontal mount traverse several grains in most orientations
and produce strongly expressed artifacts, whereas data acquisition in a vertical position
significantly decreases the number of rays that pass through more than one grain. Therefore,
particularly for samples with highly attenuating phases, we recommend scanning mounts in a
vertical position to reduce noise and improve reproducibility. A tilted orientation can achieve



similar results but makes data cropping more difficult. Scanning mounts horizontally is another,
more common option that may be suitable depending on the phase of interest.

The size and arrangement of the grains on the mount also had an influence on the

observed grayscale values and their RSDs. We tested these effects with a grain mount (Mount C)
composed of only shards of known standards (apatite, zircon, and titanite). For a vertical scan,
the horizontal position did not have an observable effect on the measured grayscale values of
grains (Fig. 8a) but the vertical position did have a significant effect, with grayscale values
decreasing downwards (Fig. 9b). This effect was observed for both apatite and zircon. Titanite
showed an even greater dependence on the vertical position, but this trend was exaggerated by
the predominance of smaller shards in the top row and larger ones in the bottom row of the
mount. These spatial effects are likely caused by the inhomogeneity of the total X-ray
attenuation at any height above the sample holder due to clustering of grains at certain heights.
These spatial effects can be minimized by distributing known standards throughout the grain
mount and normalizing sample grain measurements by the closest standard, and by avoiding
lines or grid shapes when placing grains.

We observed a general trend of decreasing grayscale values with increasing grain size for

the set of all grains of this mount (Fig. 9c). This trend can be explained by beam hardening (see
Hanna and Ketcham, 2017), which results from the preferential attenuation of low-energy parts
of the X-ray spectrum by highly attenuating material. This effect makes the center of highly
attenuating regions appear darker. This artifact can lower the observed average grayscale value
of a grain, producing measurements that are not solely related to the attenuation coefficient of a
phase. This can be counteracted by choosing standard grains/shards that are matched in size to
the unknown sample grains. If beam hardening occurs, it will affect all grains equally, thereby
allowing for a direct, unbiased comparison of the average grayscale values of sample grains and
standards.

The geometric effect discussed above can change the average observed grayscale values

of grains by 5-10%. Even with these effects, apatite can still be distinguished from zircon due to
their large relative difference in X-ray attenuation. However, precautions should be taken when
distinguishing apatite from titanite, which displays a much lower relative contrast (see Figs. 4, 5,
6), to ensure that data quality is high and phase identification is robust and unique.

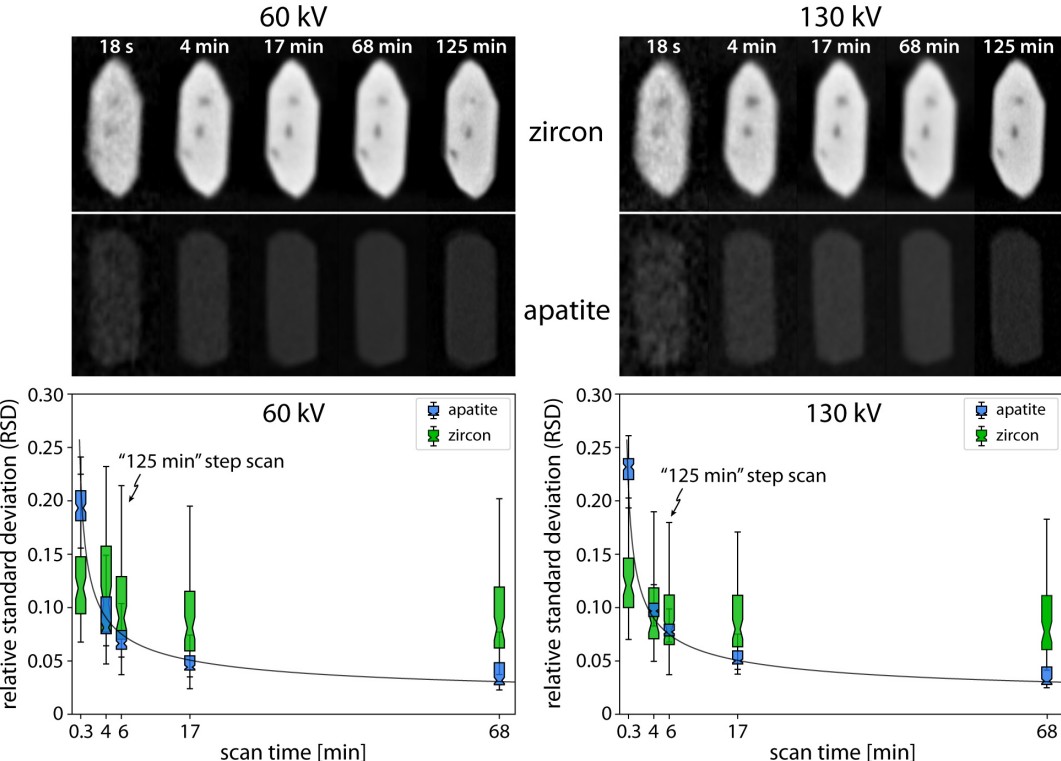


Figure 7: Slices of selected grains (top) and grayscale relative standard deviations (RSDs) of all

analyzed apatite and zircon grains (bottom) at different scan times for 60 kV and 130 kV scans.

Slices are given at the same contrast settings, showing the difference in grayscale value between

apatite and zircon. Scans of 18 s, 4 min, 17 min, and 68 min are processed at a reduced

resolution (5.7 µm) whereas 125 min scans are processed at full resolution (2.1 µm). Image

quality and signal-to-noise ratio improve with longer scan times, and graphs of $1/\sqrt{n}$-functions

are given for reference (gray lines). For our instrumental and scan parameters, we did not see

significant improvements in signal-to-noise ratio past 17 min.




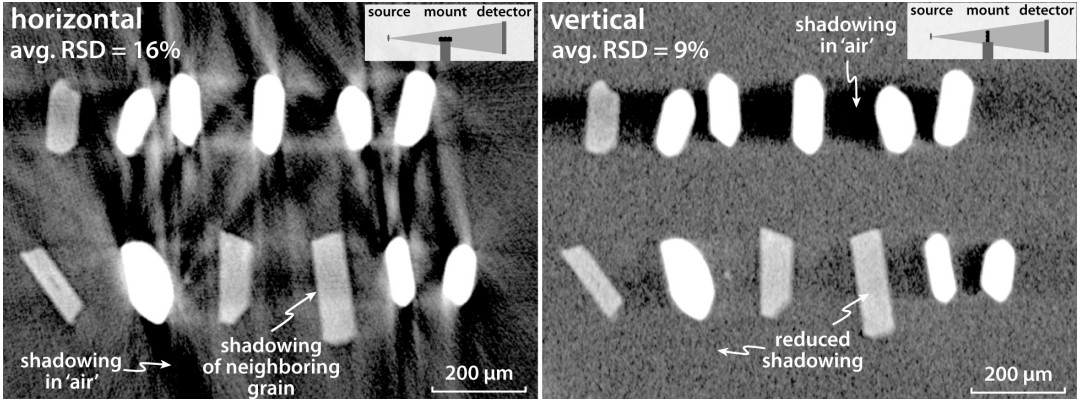


Figure 8: Slices of horizontal and vertical scans of the same grain mount show the reduction of
artifacts for the vertical scan position relative to the horizontal scan position. Highly attenuating
zircon (bright) grains produce shadowing artifacts that overlap with apatite (less bright) grains,
altering the overall grayscale value measured in the apatite grains. Some shadowing still occurs
in the vertical position but is much reduced relative to the horizontal position. This is reflected in
the relative standard deviation (RSD) of the grayscale value within each set of grains. The
arrangement of grains in a geometric pattern leads to the amplification of artifacts. Note:
Photographs have increased contrast to highlight the differences in artifacts.


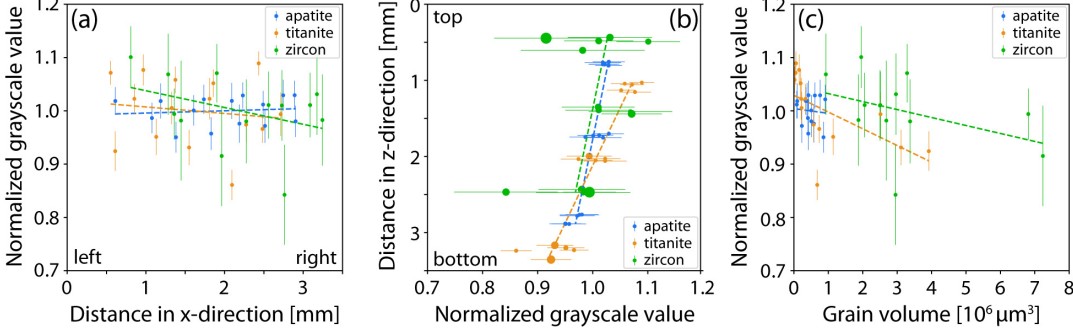


Figure 9: Plots showing the effect of spatial parameters on the grayscale values of the grains on
Mount C, which contains shards of known apatite, titanite, and zircon crystals (see Fig. 2). The
measured grayscale values have been normalized by the average of all grains of that mineral.
Linear regressions (dashed lines) show approximate trends. (a) There is no systematic variation
of normalized grayscale values with horizontal distance (x-direction) of grain placement on the





mount. (b) The normalized grayscale values of all mineral grains show a dependence on vertical
distance (z-direction) on the mount. The trends of decreasing brightness from top to bottom are
roughly parallel for apatite and zircon, with around 5% total variation. Titanite shows larger
grayscale variations (~10%), which are partly due to variations in the volume of grains (size of
symbol correlates with volume). Larger grains are preferentially located at the bottom of the
mount, thereby amplifying this trend. (c) Grains of larger volume have lower grayscale values,
likely due to the effects of beam hardening.

**3.5 Recommended procedures for microCT phase identification for geo- and**

**thermochronology**

Based on the calibrations above, we have developed a workflow for the identification of
apatite and zircon grains in grain mounts for geochronology using microCT. The methodology
described here has the potential to eliminate the need for highly toxic heavy liquids (MEI and
bromoform), reduce time spent picking grains, and curtail misidentification of apatite and zircon
in geo- and thermochronological analyses. Instead, this enables the use of less toxic heavy
liquids (LST, LMT, SPT) that produce mixed apatite and zircon separates and users can quickly
pick suitable-looking grains without close visual inspection and appraisal of interference colors,
crystal shape, etc. If the objective is to simply distinguish between apatite and zircon, then
reconstructed grayscale slices of rapidly acquired (~10-20 min) microCT data can be used to
visually identify the mineral phase of each grain, requiring little technical training and using
freely available software such as ImageJ (Schneider et al., 2012). For a more quantitative record
or if the separation of phases with a small, weak density contrast (such as apatite and titanite) is
required, grains can be segmented with more specialized software (such as Dragonfly, which
offers free academic licenses), and average grayscale values can be extracted for each grain. For
many geochronological applications, both apatite and zircon are desirable target phases.
Therefore, this method can be used to screen for both minerals at the same time. For the
detection of inclusions and the 3-dimensional measurement of grain geometry, this method can
be used with microCT scans with longer scan times (~2 h), which can be processed to yield a
better spatial resolution.
We found that using clear plastic slides (thickness ~0.5 mm) as a base for grain mounts
provided the necessary rigid support to hold the grain mounts in place while handling during





microCT scanning. These plastic slides have a similar refractive index to glass and can be easily
cut with scissors or other implements. Exact mount shapes (circles, squares, rectangles) depend
on the scanner set-up. Generally, the goal is to maximize the grain mount surface area to fit a
large number of grains on a single mount. As mentioned, double-sided adhesive tape is strong
enough to secure mineral grains, even in vertical scans, but different tapes can vary in terms of
clarity and glue thickness.
Unknown mineral grains can be picked from a separate and placed directly onto the grain
mount with tweezers or a needle. The grains should be placed onto the adhesive tape firmly
enough to ensure that enough surface area of the grain is in contact with the tape, but not so
firmly that the grain breaks. We recommend strategically distributing the unknown grains in such
a way that any individual grain can be easily identified after microCT for further analysis. Grains
should be spaced at least one grain length apart to minimize the effect of artifacts from highly
attenuation phases. Forming lines or a grid of grains should be avoided since these shapes tend to
amplify artifacts. Known mineral standards of expected phases should be included on every grain
mount. They can be shards of larger crystals or mineral grains that have been identified by an
independent method, such as through micro-Raman spectroscopy. These standard grains should
broadly match the grain sizes of the unknowns and be distributed throughout the grain mount in
the same way as the unknowns to account for any spatial variation in X-ray attenuation. In some
cases, the mineral standard can also be used as the age standard for further analysis (e.g.,
Durango apatite).
Vertical grain mount scans produce overall better results by reducing microCT artifacts
(see Fig. 7). However, horizontal scans are likely sufficient in many applications, such as
distinguishing apatite and zircon, and allow multiple grain mounts to be stacked on top of the
sample holder. This allows 4-times the number of grains in a single scan (up to 400 grains). The
resulting file sizes will be bigger, but the scan time is the same.
**3.6 Benefits of microCT in geo- and thermochronology**
Here we present a rapid method for identifying apatite and/or zircon crystals in separates
using microCT as a pre-screening technique. This can serve several purposes depending on the
goal of the research. First, it can reduce the misidentification of minerals prior to costly and time-





intensive analyses. In the case of precious or low-yield samples, reducing human error is
especially important.

The 3D grain-specific measurements acquired during the micro-CT scan provide added

value to (U-Th)/He thermochronology research where grain shapes are used to calculate Ft
corrections and directly impact age calculations. These corrections typically assume a mineral
grain geometry and use 2D grain measurements (e.g., Farley et al., 1996). More recent work has
used microCT to calculate 3D Ft and/or validate 2D Ft measurements (Evans et al., 2008;
Glotzbach et al., 2019; Cooperdock et al., 2019). The method presented here yields data that can
be directly used with the Blob3D software for 3D Ft calculation, or provide more precise grain-
specific surface area and volume measurements for calculating Ft by hand.

For detrital geochronology, the microCT pre-screening method described here can be

used to identify mineral phases regardless of grain geometry, thereby enabling the use of grains
with less-than-ideal geometries. Since apatite and zircon are mainly picked under a binocular
microscope based on their grain shape, sub-euhedral or broken crystals, which typically represent
the bulk of the crystals in a given separate, are often not chosen for further analysis. This can
present a problem for samples with low yields or bias the results to grains of specific
morphologies (i.e., histories or age populations).

Furthermore, this method can be expanded beyond apatite, zircon, and titanite. For

example, we did not analyze monazite or rutile in this study. However, based on the MuCalc
modeling and the characteristics of the microCT scans analyzed here, monazite and rutile should
be distinguishable from apatite, zircon, and titanite at X-ray energies below ~200 keV, with a
greater distinction between these phases at lower X-ray energies. The separation of common
detrital minerals, such as apatite, zircon, titanite, monazite, and rutile in a grain mount, crushate,
or rock sample could also be used for detrital heavy mineral analysis.
**4 Conclusions**

We show that microCT pre-screening of grains picked from separates can be used to

unequivocally distinguish apatite and zircon, and to distinguish apatite and zircon from other
phases, such as titanite, with a degree of certainty. Normalizing grayscale values of grains from
microCT volumes by the average value of a known zircon standard accounted for differences in
experimental setup, instrument performance, and processing from one mount to the next. The

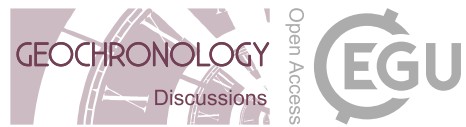

remaining observed variation of grayscale values within and between grains is likely due to
grain-specific natural variability of material parameters, such as crystal damage and elemental
substitution.
We recommend the following best practices for future studies:
● Mineral standards for normalization should be matched in size to the unknown samples to
account for the effect of beam hardening.
● Standards should be distributed throughout the mount, and sample grains should be
normalized by the closest standard grain to minimize minor spatial effects.
● The mount should be tilted vertically for the microCT data acquisition to reduce the
effect of shadowing from neighboring grains. MicroCT instrument geometries other than
the one used here might require different mount orientations.
● For the particular microCT instrument used here, the signal-to-noise ratio did not
improve significantly past 17 min for continuous scans. A step scan of about 2 h (50 min
counting time) was sufficient to produce high-resolution data with a usable signal-to-
noise ratio.
MicroCT scans that are set up according to the recommendations are a robust method to
distinguish between apatite and zircon in mounts of selected grains. This offers a possible
alternative to separating apatite from zircon using highly-toxic MEI. Grains can be picked
directly from separates that have undergone a density separation with non-toxic LST, LMT, or
SPT, which is a less laborious and safer process. As an additional benefit, the data acquired in
this process can also be used to screen the sample grains for fluid and mineral inclusions and to
model alpha-ejection and -implantation corrections for (U-Th)/He dating (Evans et al., 2008;
Cooperdock et al., 2019).
**Data availability**
Reconstructed microCT volumes for all mounts, X-ray energies, and scan times are stored at the
USCHelium Lab and are available on request.
**Author contribution**
EHGC and FH conceptualized the study and experimental design with input from AT; AC
collected FCT samples; FH, RMC, and AC prepared samples and collected data; all co-authors



contributed to data interpretation; FH and RMC prepared figures; EHGC and FH prepared and
edited the manuscript draft with input from RMC, AC, AT, and AJC.
**Competing interests**
AT is a representative for Rigaku Americas Corporation, the company which manufactured the
microCT instrument used in this study.
**Acknowledgments**
We thank Justine Grabiec and Alexia Rojas for help with mineral separation; Danny Stockli and
members of the UTChron laboratory for providing Fish Canyon Tuff and Durango samples; and
Ken Farley for providing the Caltech Durango sample. We thank Kalin McDannell, Paul
O'Sullivan, and Ryan Ickert for useful discussions about heavy liquids safety, and James Metcalf
for FCT sampling information. We also thank Alan Gregorski and Aaron Alke for help sampling
the FCT.

**Financial support**
This work was supported by Emily H. G. Cooperdock's University of Southern California start-
up funds and a Major Support Funding Grant from the Women in Science and Engineering
(WiSE) at the University of Southern California.

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
