# Peer review of "Technical note: Rapid phase identification of apatite and zircon grains for geochronology"

_Geochronology, 2022_

## Author Response (AR1)

Response to referee comment by Anonymous Referee #1

**We thank Referee #1 for their review and constructive suggestions. We have responded to all points and made changes according to the referee's suggestions. Author responses below are given in bold.**

I appreciate the detailed and careful work that the authors have done developing and optimizing a protocol for microCT imaging of zircon and apatite. I believe that there is great, untapped potential for geochronological applications of microCT imaging of loose grains, grain mounts, and whole rock samples. Others have shown that microCT imaging can be used to quantify grain size, characterize crystal morphology, estimate surface-to-volume ratios for alpha-ejection corrections, and to identify inclusions and fractures in minerals. As such, microCT imaging could well become an important, routine tool for zircon and apatite chronology like it has become for (U-Th)/He dating of magnetite and other opaque oxides.

I also appreciate that the authors highlight the health risks of toxic heavy liquids like methylene iodide and bromoform. I agree that whenever possible, use of these chemicals should be avoided. Non-toxic heavy liquids such as LST, panning, and Wilfley tables are effective methods for separating out less dense minerals. However, as stated by the authors these methods are not capable of – or are not always terribly effective at – separating apatite from zircon.

While I believe that the work presented here represents an important contribution to the field of geochronology, I question the practicality of using microCT as a routine tool for mineral identification. Mineral identification is indeed a challenge in geochronology; geochronological opportunities are missed if less routinely dated minerals like baddeleyite or xenotime are not accurately identified in mineral separates. Apatite and zircon, however, are routinely encountered and have distinct crystal habits, optical properties, and differing solubilities. In the case of mineral identification challenges, other tools like Raman spectroscopy and EDS analysis have databases at hand that can identify mineral phases more directly as compared to relying on microCT density contrast. Further, not many geochronology labs are equipped with in-house microCT scanners. That said, having another analytical tool available to help with mineral identification – in the event that Raman and EDS are not available, and optical identification has truly failed – is useful.

**We agree with the points raised by the referee. Although microCT is not readily available to everyone in the community, there are many facilities that offer paid microCT scanning services. The data acquisition periods needed for our proposed method are relatively short (<2 h), which can be done economically. At current prices (~$100 per hour), this adds little more than the costs of consumables and paying a worker to perform mineral separation. If mounts are stacked, thousands of grains could be analyzed in a single scan, thereby optimizing scan time and costs. MicroCT scanning offers the opportunity to quickly obtain data for a large number of grains at once, for which many individual analyses would be required with EDS or Raman spectroscopy. The method discussed here can be used for efficient pre-screening if an in-house microCT instrument is available, or it can be used to complement other analytical techniques.**

**We have added text about the availability of microCT and the time investment of the procedure in the discussion section of the manuscript.**

In my personal experience, the challenge separating zircon from apatite in LST dense fractions that have not undergone MEI separation is often a problem of relative abundance. Some rocks have significantly

more apatite than zircon. Identifying a small number of zircon crystals in an ocean of apatite can mean significant time spent at the picking scope. While spending more time at the microscope is safer than using MEI, I doubt that using microCT imaging to aid in mineral identification helps to save time.

**Since zircon grains stand out very clearly compared to apatite grains, microCT scanning could be utilized for the exact scenario described here. Thousands of grains could be quickly spread out on a piece of adhesive tape. Even in short scans (~2 min) zircon grains would be immediately apparent in the scan and could then be picked from the mount. Alternatively, individual grains from such a separate could be quickly picked without close visual inspection and then analyzed with microCT to identify zircons.**

**We added text in section 3.5 that describes how this microCT method may be useful for detrital zircon LA-ICP-MS for detecting zircon and limiting the bias introduced by selecting zircon based on grain morphology alone.**

16-18: Is it really that challenging to distinguish between apatite and zircon? The two minerals have very different crystal morphologies, optical properties, and acid solubilities.

**For perfectly preserved crystal morphologies, the distinction between apatite and zircon is relatively straight-forward. However, not all samples and processing methods yield many ideal grains. Crystal morphology and optical properties can be hard to assess in less-than-ideal grains, such as those with broken tips or physically abraded grains from detrital samples. We have revised the introduction and other sections of the manuscript to more clearly state the conditions when it may be more difficult to identify apatite from zircon.**

36: Characterization of crystal shape, size, and inclusion content seems a stronger motivation.

**We will emphasize these aspects of the microCT analysis throughout the manuscript. Conversely, phase identification/confirmation can be an additional benefit of pre-screening apatite and zircon samples with microCT for crystal-shape analysis and could be integrated into existing protocols. In order to detect inclusions by microCT, especially in a (semi-)automated way, a baseline for the host phase must be established. Segmentation and calculating the average brightness value of each grain relative to reference grains, as discussed in this study, will also facilitate this process.**

57: I appreciate the discussion about health risks. I feel that most users don't know much beyond the fact that these chemicals are "toxic" or carcinogenic.

**Indeed, it was interesting to compare our own impressions with the published literature.**

104: I would think that mounting and scanning grains likely adds additional time and cost to the process as well – not many labs have their own CT scanner.

**Making mounts adds very little time to the process, and grains would have to be picked from a separate in any case. A grain mount (plastic + adhesive tape) can be assembled in minutes and grains are places on it instead of in a container/tube.**

**We add text in section 3.5 that describes the time requirements for this procedure, and the pros and cons from more traditional picking procedures. Ultimately, use will depend on instrument availability, cost, and whether the sample needs extra validation.**

160: It would be useful/interesting to demonstrate how often trained graduate students actually misidentify zircon and apatite. It would help to justify the study's stated motivation.

**This would be an interesting study. At this time, we have not systematically documented this in a way we feel comfortable adding to the manuscript (i.e., it would not be anonymous). Ours is a new lab with new users and we have experienced students misidentifying grains. We also know from our network of thermochronology colleagues that there are various "tricks" everyone uses to address this issue (mostly Raman or SEM-EDS). This gives us confidence that it's a more common occurrence than is often openly discussed, though we acknowledge it is going to be a function of user experience and sample type.**

291-294: If a worker has already invested the time in preparing a grain mount for microCT imaging, why not simply go for the longer scan that will yield additional, more useful information about grain morphology, inclusions, ect?

**The decision would depend on constraints around allotted instrument time, budget, and/or whether information on morphology is required. Short scans can be useful to quickly verify whether the right grains have been identified during the picking process. This is especially helpful for users who are being trained to pick grains since it provides immediate feedback (assuming that a microCT instrument is available). Short scans can also be used to pre-screen larger separates from which a subset of grains will be picked based on their microCT characteristics. This subset of grains can then be scanned for longer periods of time to obtain information on inclusions and grain shape. We determined the minimum conditions for obtaining useful data. Any increase in scanning time from that point on will yield better data. The exact scanning parameters will depend on the instrument, the use of the data, and the budget.**

Figure 3: Great figure. Very useful for predicting which minerals CT scans may be useful for. Maybe include other minerals of geochronological interest? How do zircon and apatite compare to magnetite and other oxide minerals that are now the focus of many (U-Th)/He studies?

**The phases included in this figure are those which could theoretically be mistaken for apatite or zircon, at least in grains with sub-optimal morphology. Given that apatite and zircon are the focus of this manuscript, we prefer to leave the figure as-is. We are currently preparing another manuscript that includes Fe-oxides and other phases used in chronology studies where such a figure will be more relevant.**

Figure 7: As someone new to microCT imaging, this is also a great figure illustrating how different scan conditions affect image quality.

**Glad to hear it!**

Response to referee comment by Anonymous Referee #2

**We thank Referee #2 for their review and suggestions to improve the manuscript. We have responded to all points and made changes according to the referee's suggestions. Author responses below are given in bold.**

This manuscript presents a detailed description of a microCT method for distinguishing apatite from zircon. The authors clearly lay out an optimized methodology, along with excellent figures that illustrate various aspects of the data and measurements. Besides some suggestions that I provide below for clarification, I have no issue with this manuscript being published. I appreciate the authors' efforts to develop another way to use microCT together with geochronology methods. However, I'm skeptical this

method will be widely adopted for distinguishing apatite from zircon. The authors might consider shifting the manuscript emphasis in places, as suggested below, to potentially make this contribution more impactful.

In my view the problem being addressed (distinguishing apatite from zircon in mineral separates) is greatly overstated. In the vast majority of circumstances, it isn't challenging to distinguish apatite from zircon under the microscope after separation with LST. These minerals are distinct in morphology, relief, and other properties. Even for newbies, after a few hours of getting one's eyes calibrated at the microscope, it is not particularly difficult to distinguish these mineral phases. It arguably becomes more important for detrital mineral suites – it may be effective for the authors to specifically emphasize this challenge, rather than trying to argue that this is a routine problem when it really isn't and most reading this paper will know this.

Agreed that performing mineral separation with toxic chemicals is undesirable, but again this strikes me as overstated given that one can alternatively use LST and then i.d. the minerals typically without too much trouble.

**We did not intend to say that distinguishing between apatite and zircon is impossible or a particularly difficult task in many sample separates. However, there can be some challenges in distinguishing small grains, grains with broken tips, and abraded detrital grains. Therefore, there is a possibility of individual grains being misidentified, which will vary based on the experience of the person picking and the sample type. In our own lab we experienced misidentified grains, and we have heard similar stories from most other labs. We propose that microCT scanning can be an optional pre-screening or validation method which can be implemented in routine sample preparation procedures. We have clarified our phrasing and motivation throughout the manuscript, particularly in the Introduction.**

Although not emphasized in the abstract or introduction, elsewhere in the paper the authors highlight attempting to distinguish apatite from titanite with microCT. These phases are even easier to differentiate than apatite and zircon, with titanite typically coming off at a more magnetic level on the frantz than apatite. It may be better to eliminate this comparison in this paper entirely.

**Mineral separation techniques are not 100% effective, so any separate could potentially contain titanite (such as the separates in this study), which could be hard to distinguish from apatite in grains with poor morphology. We included titanite in the analyses to highlight some of the potential challenges with using microCT (e.g., phases with similar density), and we therefore think it is worth keeping the data in place.**

It would be helpful to provide some estimate of the total time required per grain (including mineral selection, mount making, analysis, data reduction) to 1) use the proposed microCT method to distinguish different mineral phases and 2) additionally identify inclusions and acquire grain geometry information.

**The total time required will depend on the level of training and familiarity with microCT data analysis as well as computer hardware. The time investment comes in the beginning as one is learning the segmentation software. Once a user is familiar with the software functions, the time investment is on the order of a couple hours maximum if they want to calculate grain-specific surface areas and volume for 3D Ft correction or grain mass estimates.**

**1) Distinguishing between apatite and zircon (as long as no other phases are present) can be done visually by viewing the microCT data in an appropriate software, which only takes a few minutes. Segmentation and average grayscale value computation can be done in a few steps in most microCT**

**software packages, such as Dragonfly, and can be (semi-)automated, requiring 5-15 min to be completed. This time is per acquired volume, so it could either apply to a single layer (as reported here), or to a stack of layers (containing hundreds or thousands of grains). 2) We have identified inclusions in grains by inspecting successive grayscale slices through each grain. Identifying inclusions and classifying individual crystals this way takes <1 min per grain. Calculating grain-specific volumes and surface area are done in batch and can be (semi)-automated like the segmentation computation mentioned above in 15 min or so using Dragonfly software.**

**In the end, this is not significantly more time than is required to measure via 2D methods on a microscope (which does not need to be done if one uses CT), and has the added benefit of being more accurate for odd shaped grains.**

**We add this discussion in section 3.5 and 3.6.**

Is the microCT method for mineral i.d. faster than alternative analytical methods that could be used to identify these phases? For example, in my experience, mineral identification using an EDS system on an electron microprobe or SEM requires only a few minutes to place individual crystals on carbon tape and then seconds per grain for EDS identification. This seems faster than the microCT method described here, and EDS systems are more common and therefore more accessible than microCT systems. If this is incorrect, then it would be helpful to clarify this in the paper.

**From our experience, a single-layer grain mount can be assembled as part of the picking process, and therefore requires little additional time to prepare, which is similar to EDS. MicroCT volumes show information for all grains simultaneously. If pure phase identification is required, it can be done visually, which only takes seconds. This is faster than manually placing dozens of Raman or EDS spots. However, these types of analyses can yield information about composition, which might be preferable in some situations. Overall, microCT can be used as the main pre-screening method or it can be combined with other methods, such as the ones mentioned, for a more thorough sample characterization.**

If one is going to the trouble of making the mount, then why not use the longer scan times to acquire the additional information about inclusions and grain geometry? This strikes me as a more compelling reason to use this method, and could be emphasized more strongly as a motivation in the paper. Or perhaps this could become the paper's primary motivation.

**Other papers have described the procedure for grain geometry analysis and inclusion mapping (e.g., Evans et al, 2008; Cooperdock et al., 2019; Glotzbach et al., 2019). Additionally, we have a manuscript focusing on using microCT to detect inclusions in various minerals, including optimizing scan parameters and analysis procedures is in preparation. Phase identification can be one motivating factor for pre-screening apatite and zircon grains with microCT. Phase identification, as described here, can be integrated into the routine microCT data analysis for pre-screening procedures.**

Lines 33-43: Suggest revising the second sentence. The characterization of the crystal shape does not matter for U-Pb and fission-track, unlike what is implied by the structure of these two opening sentences.

**We have revised to sentence to be clear these are important factors for (U-Th)/He.**